# High mortality rates and long-term complications in children with infectious brainstem encephalitis: A study of sixteen cases

Yuanyuan Zhou[1☯], Yi Zhu[1☯], Lingfeng Cao[2], Yingyan Shi[3], Jun Shen[1,4]*

1 Department of Infectious Disease, National Children's Medical Center, Children's Hospital of Fudan University, Shanghai, China, 2 Department of Virology, National Children's Medical Center, Children's Hospital of Fudan University, Shanghai, China, 3 Department of Radiology, National Children's Medical Center, Children's Hospital of Fudan University, Shanghai, China, 4 Department of Pediatrics, Children's Hospital of Fudan University At Qidong, Nantong, China

☯ These authors contributed equally to this work.
* shenjun_@fudan.edu.cn

**Data Availability Statement:** Within the manuscript itself.

## Abstract

### Objective

Brainstem encephalitis (BE) can cause sudden death in children. Fewer studies have been conducted on the incidence, clinical manifestations, pathogens and post-infectious sequelae of pediatric infectious BE.

### Methods

Pediatric patients diagnosed with BE in our Medical Center from 01 January 2015 to 31 July 2024 were retrospectively reviewed. The clinical data of these children were obtained from the hospital's medical database on 15 August 2024. The number of outpatient and inpatient patients at our Medical Center during that period were provided by the hospital data center. Data analysis was conducted using Excel 2019.

### Results

A total of twenty-eight cases were diagnosed with BE in our National Children's Medical Center over the past decade. Among them, 57.1% (16/28) cases were diagnosed with infectious BE. The incidence of infectious BE was estimated to be 16 cases per 30 million outpatient visits and 13 cases per 500,000 hospitalized patients. Fever, consciousness disorders and seizures were observed in 75.0% (12/16), 68.8% (11/16) and 62.5% (10/16) of the cases, respectively. Among them, 31.3% (5/16) cases were diagnosed as human enterovirus infections, 12.5% (2/16) cases were confirmed to be influenza B virus infections, while one case each was diagnosed with herpes simplex virus 1 and human herpesvirus 6 infection. The mortality rate during hospitalization was 12.5% (2/16). Among the surviving patients, 50.0% (7/14) of them had follow-up records, 85.7% (6/7) of the survivors suffered from sequelae such as motor disorders.

**Funding:** This study was supported by The National Key Research and Development Program of China (2022YFC2704901), the Project of Invigorating Health Care through Science, Technology and Education (Nantong Municipal Key Medical Discipline), and Beijing Natural Science Foundation (L242052). The funders had no role in study design, data collection and analysis, decision to publish, or preparation of the manuscript.

**Competing interests:** NO authors have competing interests

## Conclusion

Fever, consciousness disorders and seizures were the major clinical manifestations in patients with infectious BE visited our Medical Center. These rare cases exhibited a notably high mortality rate and a significant frequency of long-term complications.

## Introduction

Infectious brainstem encephalitis (BE) is frequently characterized by its high mortality rate and susceptible to long-term sequelae, which viruses are the primary causative agents [1–6]. Early identification of infectious BE and implementation of targeted antimicrobial therapies are critical for optimizing patient outcomes in clinical practice [3, 4]. Common etiological diagnostic methods for infectious BE include multiple PCR tests and metagenomic next-generation sequencing (mNGS) to detect potential etiologies, including but not limited to herpes simplex virus (HSV), human enterovirus virus (HEV), and human parechovirus [7–10]. There are few reports about the clinical characteristics and prognosis of children with infectious BE in China, especially during and after the COVID-19 pandemic [2]. Currently, there is a lack of treatment guidelines available for clinical pediatricians to guide the management of infectious BE, including the utilization and supported evidence of intravenous immunoglobulin (IVIG) as well as its potential benefits, which poses a challenge in delivering specialized care.

Here, we conducted a comprehensive analysis of the clinical data pertaining to pediatric patients diagnosed with infectious BE in our Medical Center over the past decade. The aim of this research is to provide additional evidence for the identification of children with infectious BE, and enhance our comprehension of the long-term consequences associated with infectious BE in pediatric patients.

## Materials and methods

### Patients

This study included pediatric cases diagnosed with infectious BE in our National Children's Medical Center from 01 January 2015 to 31 July 2024. The criteria for diagnosing infectious BE include: (1) presenting with clinical manifestations such as fever, seizures, altered consciousness, cranial nerve dysfunction, respiratory and/or cardiac arrest, and exhibiting abnormal signal shadows in the brainstem on cranial magnetic resonance imaging (MRI); (2) displaying typical infectious BE symptoms like respiratory and/or cardiac arrest when cranial MRI is not performed or showing no abnormalities [3]. Cases with non-infectious BE conditions, such as immune-mediated disorders, genetic metabolic disorders, tumors, trauma and toxic exposures were excluded.

The clinical data of pediatric cases with BE was retrieved from the hospital medical database on 15 August 2024, encompassing initial age at diagnosis, gender, clinical presentations, pathogen identification (human enterovirus [HEV], influenza virus [FLU], herpes simplex virus [HSV], cytomegalovirus, Epstein-Barr virus, bacterial and fungal culture, parasites), cranial MRI and electroencephalogram (EEG) examinations, treatment modalities employed as well as subsequent follow-up procedures. The number of outpatient and inpatient patients at our Medical Center from January 2015 to July 2024 was provided by the hospital data center on15 August 2024.

## Statistical analysis

The continuous variables that followed a normal distribution were reported as mean ± standard deviation, while those deviating from normality were presented as the median and interquartile range (25%, 75%). Categorical variables were expressed as frequencies and percentages, including the proportion of infectious BE among outpatient and inpatient children during January 2015 to July 2024. Excel 2019 was used for data analysis.

## Ethical statement

This study was conducted according to the guidelines established in the Declaration of Helsinki, and all procedures involving human subjects/patients were approved by the Ethics Committee of Children's Hospital of Fudan University (No. 2024–33). Written informed consent was obtained from all from pediatric patient's legal guardians, and confidentiality and anonymity of their provided information were ensured. All data were collected anonymously and handled confidentially.

# Results

## Demographic information

A total of twenty-eight cases were diagnosed with BE. Among them, 57.1% (16/28) of cases were diagnosed with infectious BE, while the remaining 42.9% (12/28) of cases were non-infectious including seven cases of immune-mediated BE, two cases of genetic metabolic BE, two cases of tumor-related BE, and one case of toxic induced BE.

Among the sixteen cases of infectious BE, there were an equal number of boys and girls (eight each). The median age at diagnosis was 4.0 (1.4, 6.0) years, with the age range spanning from one year and two months to eight years. Apart from one case diagnosed with congenital heart disease, none of the sixteen patients had any pre-existing medical conditions prior to contracting the infectious BE. Additionally, there were no case had immune inhibitor exposure.

Thirteen cases were admitted for hospitalization during the acute phase of infectious BE. Three cases were treated as outpatient who referred from other hospitals. The number of infectious BE cases in the years 2017, 2018, 2020, 2021, and from January to July 2024 were four, two, four, two, one and two respectively. No infectious BE cases were identified in the years 2015, 2016 and 2019 (Table 1). Meanwhile, our medical center had observed approximately 30 million outpatient visits and 500,000 hospital admissions during that period. Consequently, the incidence of infectious BE among outpatient cases was estimated to 16 cases per 30 million visits, while among hospitalized cases it was roughly 13 cases per 500,000 admissions.

## Clinical manifestations

Among the cases with infectious BE, 75.0% (12/16), 68.8% (11/16), and 62.5% (10/16) of the cases presented with fever, consciousness disorders (including five cases with coma and three of them with deep coma), and convulsions respectively. Gastrointestinal symptoms were observed in 62.5% (10/16) of the cases, with six cases experiencing vomiting and four cases experiencing diarrhea; two cases had both vomiting and diarrhea simultaneously. There were 31.3% (5/16) of the cases exhibited cranial nerve involvement, with two cases presenting swallowing disorders, two cases demonstrating dysarthria, and one case displaying oculomotor disorders. Furthermore, four cases experienced dyspnea, while four others suffered from respiratory arrest (three of which were accompanied by cardiac arrest). Additionally, three cases displayed limb weakness or paralysis, and two cases exhibited skin rashes.

**Table 1. Clinical details of the sixteen infectious brainstem encephalitis cases.**

| Cases | Date at diagnosis (Y/M) | Gender | Age at diagnosis (y/m) | Clinical manifestation | Etiology | CSF test * | Cranial imaging | EEG | IVIG giving | Follow-up span (d) | Sequelae or death |
|---|---|---|---|---|---|---|---|---|---|---|---|
| 1 | 2017/03 | Female | 8y | No fever, Outpatient recovery period | Unknow | Unknow | MR: Brainstem abnormal signal | Unknow | No | 408 | Learning disability |
| 2 | 2017/03 | Male | 2y11m | Fever, consciousness disorder, rash, dyspnea, swallowing disorder | HEV | WBC (20), Protein (184), Glu (4.1) | MR: Brainstem abnormal signal | Unknow | Yes | 222 | No |
| 3 | 2017/07 | Male | 1y5m | Fever, convulsion, consciousness disorder, rash, dyspnea | HEV | Unknow | MR: Brainstem abnormal signal | Unknow | Yes | 102 | Decreased muscle strength |
| 4 | 2017/11 | Male | 1y2m | Fever, vomiting, diarrhea, convulsion | Negative | WBC (2), Protein (387.9), Glu (5.7) | MR: Brainstem abnormal signal | Encephalitis-like slow waves and low voltage | Yes | 1737 | Language and motor disorders, secondary epilepsy |
| 5 | 2018/01 | Female | 4y6m | Fever, vomiting, convulsion, consciousness disorder | FLU-B | WBC (0), Protein (208.3), Glu (3.7) | MR: Brainstem abnormal signal | Encephalitis-like slow waves and low voltage | No | 63 | Language and motor disorders |
| 6 | 2018/07 | Male | 1y2m | HFMD-BE outpatient recovery period (fever, vomiting, convulsion, dyspnea) | HEV | Unknow | CT: Brainstem abnormal signal | Unknow | Unknow | 2065 | Secondary epilepsy |
| 7 | 2020/01 | Male | 6y | Fever, convulsion, consciousness disorder,dyspnea | FLU-B | WBC (6), Protein (436.4), Glu (6) | MR: Brainstem abnormal signal | Encephalitis-like slow waves and low voltage | No | Lost | Unknow |
| 8 | 2020/10 | Female | 6y | Fever, consciousness disorders, limb weakness, oculomotor disorder | Negative | WBC (0), Protein (173.4), Glu (3.2) | MR: Brainstem abnormal signal | Encephalitis-like slow waves and low voltage | Yes | Lost | Unknow |
| 9 | 2020/10 | Female | 2y3m | Convulsion, consciousness disorder, diarrhea, limb weakness | Negative | WBC (0), Protein (219.8), Glu (4.1) | MR: Brainstem abnormal signal | Encephalitis-like slow waves and low voltage | Yes | Lost | Unknow |
| 10 | 2020/12 | Female | 1y5m | Fever, vomiting, convulsion, consciousness disorder, respiratory arrest | Negative | Undo | Undo | Encephalitis-like slow waves and low voltage | No | Not applicable | Death |
| 11 | 2021/06 | Male | 5y7m | Fever, vomiting, diarrhea, convulsion, consciousness disorder, respiratory and cardiac arrest | HEV | Undo | Undo | Encephalitis-like slow waves and low voltage | Yes | Lost | Unknow |
| 12 | 2021/12 | Male | 3y6m | Fever, vomiting, convulsion, consciousness disorder, respiratory and cardiac arrest | Negative | Undo | Undo | Encephalitis-like slow waves and low voltage | No | Lost | Unknow |

(*Continued*)

**Table 1.** (Continued)

| Cases | Date at diagnosis (Y/M) | Gender | Age at diagnosis (y/m) | Clinical manifestation | Etiology | CSF test * | Cranial imaging | EEG | IVIG giving | Follow-up span (d) | Sequelae or death |
|---|---|---|---|---|---|---|---|---|---|---|---|
| 13 | 2022/01 | Female | 6y | Convulsion, consciousness disorder, diarrhea, respiratory and cardiac arrest | Negative | WBC (9), Protein (24064.7), Glu (6.6) | CT: Brainstem abnormal signal | Encephalitis-like slow waves and low voltage | Yes | Not applicable | Death |
| 14 | 2023/07 | Female | 1y3m | No fever, HFMD-BE outpatient recovery period | HEV | Unknow | Unknow | Unknow | Unknow | Lost | Unknow |
| 15 | 2024/05 | Female | 8y | Fever, swallowing, disorder dysarthria | HSV-1 | WBC (8), Protein (446.5), Glu (3.3) | MR: Brainstem abnormal signal | Encephalitis-like slow waves and low voltage | Yes | 53 | Language disorders, swallowing disorders |
| 16 | 2024/07 | Male | 7y | Fever, consciousness disorder, dysarthria, limb weakness | HHV-6 | WBC (0), Protein (277.8), Glu (3.8) | CT: Brainstem abnormal signal | Encephalitis-like slow waves and low voltage | Yes | Lost | Unknow |

\* CSF test: WBC (reference value $<10*10^6$/L), Protein (reference value $<100$mg/dL), Glu (reference value ($\geq 2.5$mmol/L)

Abbreviations: HEV, human enterovirus; FLU-B, influenza virus B; HSV-1, herpes simplex virus 1; HHV-6, human herpesvirus 6; Glu, glucose; EEG, electroencephalogram; IVIG, intravenous immunoglobulin.

## Etiology

Seven of the sixteen infectious BE cases including three transferred from other hospitals, and four hospitalized cases, lacked lumbar puncture or cerebrospinal fluid (CSF) tests. All four hospitalized patients presented with critical conditions requiring invasive ventilation. One patient died, while the remaining three experienced brain herniation and were discharged against medical advice by their families. Conventional pathogen detection and metagenomic next-generation sequencing (mNGS) were used to detect CSF in nine cases who had undergone lumbar puncture during their hospitalization, and only two of them tested positive for HSV-1 and human herpesvirus 6 (HHV-6), respectively. The mNGS of CSF test for one HSV-1 positive case yielded positive results twice, with read values of 14694 and 292 respectively, within a six-day interval. While the HHV6 case was positive with an mNGS read value of 72.

Five cases were diagnosed as HEV-infected. Among them, three cases presented clinical symptoms consistent with hand-foot-and-mouth disease (HFMD). One case was tested positive for coxsackie virus A16-RNA in the stool sample, while another case tested positive for HEV-RNA in the stool. The presence of FLU-B infection was confirmed in two cases, one case identified with positive FLU-B-RNA in the sputum by polymerase chain reaction, and another case identified with positive FLU-B antigen in the nasopharyngeal swab. This search yielded no BE related to COVID-19.

## Cranial imaging and electroencephalogram

Twelve of the sixteen infectious BE cases underwent cranial MRI or computed tomography scan, all of which revealed abnormal signal abnormalities in the brainstem (Fig 1). The completion of cranial MRI before discharge was hindered by mechanical ventilation required throughout the hospitalization in three additional cases, as advised against by their families. However, all patients displayed symptoms of respiratory failure and/or cardiac arrest and were clinically diagnosed with infectious BE. Another patient transferred from a local hospital had

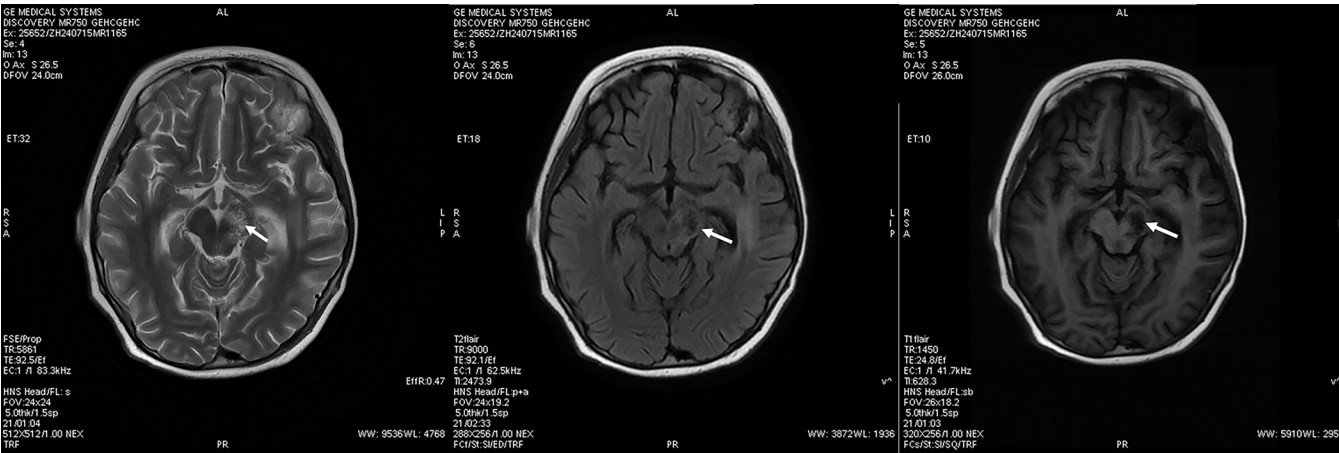

**Fig 1. Cranial MRI of a case presenting with infectious brainstem encephalitis.** The cranial MRI was conducted on the fourth day following admission for a seven-year-old boy diagnosed with HHV-6 BE, revealing a lesion site in the brainstem as indicated by the arrow.

no cranial imaging data because the BE associated with HFMD occurred five years ago. Eleven of the sixteen infectious BE cases underwent an EEG examination, all revealing encephalitis-like slow waves and low voltage on the EEG.

## Hospitalization treatment and conditions when discharged

The median length of hospital stay for the thirteen admitted cases was 24 (12, 33) days. All of the cases were admitted to the intensive care unit (ICU). Eleven cases underwent mechanical ventilation treatment, while two cases received high-flow nasal cannula oxygen therapy. Nine of the thirteen cases were admitted IVIG, whereas intravenous glucocorticoids were used in five cases. In the nine children who received IVIG experienced one fatality, whereas among the four children who did not receive IVIG, there was also one death.

When be discharged, three cases presented with consciousness disorders, two cases required mechanical ventilation due to insufficient spontaneous breathing, two cases exhibited swallowing disorders, one case experienced decreased muscle strength, one case displayed increased muscle tone, one case demonstrated psychomotor developmental delay and one case showed cognitive impairment. Two of the thirteen admitted cases died during their hospitalization.

## Follow-up

Among the eleven cases that survived and were discharged, as well as the three outpatient cases, follow-up records were available for seven cases, including two outpatient cases and five hospitalized cases. The median duration of follow-up was 222 (63, 1737) days. Six cases presented with sequelae, including one case each of secondary epilepsy, learning disabilities, decreased muscle strength, language disorders combined with swallowing disorders, language disorders combined with motor disorders, language and motor disorders combined with secondary epilepsy. The other one case with HFMD complicated by BE experienced dysphagia upon discharge and achieved a full recovery after a seven-month period.

## Discussion

Infectious BE is relatively rare in the pediatric population. Over the past decade, the estimated occurrence of encountering infectious BE among outpatient cases was 16 of every 30 million

visits, while among hospitalized cases it occurred roughly 13 of every 500,000 admissions in our Medical Center. The median age of the sixteen cases diagnosed with infectious BE was 4.0 years old. *Casas-Alba D* et al reported that the average age of forty-one children with enterovirus A71 infectious BE was approximately twenty-seven months old [4, 11]. Preschool children constitute a high-risk group, possibly due to their ongoing nervous system development at this stage.

The most common clinical presentations observed in the sixteen cases with infectious BE included fever, consciousness disorders and convulsions. Given the limited number of cases in this study, we were unable to identify any risk factors associated with infectious BE. As is well known, the brainstem plays a crucial role as the central hub of the human body. In this study, children with infectious BE exhibited evident cardiopulmonary failure, with a higher prevalence of central respiratory dysfunction as previously reported [4, 12, 13].

In these sixteen cases, the majority lacked pathogenic evidence from the CSF. HEV, FLU-B, HSV-1 and HHV-6 have been previously proven to be potential pathogens [3, 14, 15]. HEV is the primary causative agent responsible for infectious BE, with enterovirus A71 being more prevalent. Recent reports have indicated that enterovirus D68 infection can also lead to BE [11, 12]. The utilization of next-generation sequencing holds promising prospects for the pathogenic diagnosis of infectious BE [16, 17]. HSV-1 and HHV-6 was detected in the CSF from two patients respectively by mNGS in our study.

Cranial MRI and EEG examinations are crucial diagnostic methods for BE. Among the twelve cases that underwent cranial MRI, all exhibited manifestations of brainstem involvement. The MRI predominantly demonstrated hypointense signals on T1WI and hyperintense signals on T2WI [11, 18, 19]. Meanwhile, all cases that underwent EEG examinations displayed EEG manifestations resembling encephalitis.

In this study, thirteen cases were admitted to the ICU, and two cases died during hospitalization. Early administration of specific anti-pathogen treatment, IVIG, and glucocorticoid therapy may confer benefits in the early stage of infectious BE [20–22]. Nine cases received IVIG treatment while five cases underwent glucocorticoids therapy in our study. Among the two fatal cases, one case received IVIG treatment.

Children with infectious BE often experience a range of long-term complications. In this study, 56.3% of the surviving children underwent follow-up assessments, with a median duration exceeding seven months. Among these cases, 43.8% exhibited sequelae such as movement disorders, language impairments, dysphagia, reduced muscle strength, and secondary epilepsy. *Huang* et al conducted a follow-up study on sixty-three children with enterovirus A71 infectious BE and observed that 14.3% of the children exhibited persistent cognitive and movement disorders after a two-year period [5]. *Liang* et al performed cranial MRI follow-ups on children infected with enterovirus A71 or coxsackie virus A16 after four years, revealing that 57.0% of the children experiencing cardiopulmonary failure displayed abnormalities [6].

In conclusion, although infectious BE is indeed rare in pediatric clinical practice, it should still be considered as a potential diagnosis in patients presenting with fever, altered consciousness, and seizures due to its potentially fatal consequences and concerning long-term sequelae [11, 22]. Timely recommendations for brain stem MRI should be provided, taking into account the possibility of enterovirus infection.

## Author Contributions

**Conceptualization:** Yuanyuan Zhou, Yi Zhu, Lingfeng Cao, Yingyan Shi, Jun Shen.

**Data curation:** Yuanyuan Zhou, Yi Zhu, Lingfeng Cao, Yingyan Shi, Jun Shen.

**Formal analysis:** Yuanyuan Zhou, Yi Zhu, Yingyan Shi, Jun Shen.

**Funding acquisition:** Jun Shen.

**Investigation:** Yuanyuan Zhou, Yi Zhu, Lingfeng Cao, Yingyan Shi, Jun Shen.

**Methodology:** Yuanyuan Zhou, Yi Zhu, Lingfeng Cao, Yingyan Shi, Jun Shen.

**Project administration:** Jun Shen.

**Resources:** Lingfeng Cao, Yingyan Shi, Jun Shen.

**Software:** Yuanyuan Zhou, Yi Zhu, Jun Shen.

**Supervision:** Jun Shen.

**Validation:** Yuanyuan Zhou, Yi Zhu, Lingfeng Cao, Yingyan Shi, Jun Shen.

**Visualization:** Yuanyuan Zhou, Yi Zhu, Lingfeng Cao, Yingyan Shi, Jun Shen.

**Writing – original draft:** Yuanyuan Zhou, Yi Zhu, Lingfeng Cao, Yingyan Shi.

**Writing – review & editing:** Jun Shen.

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
