## [Decision Letter · Decision Letter 0]

10 Jan 2025

PONE-D-24-57941High mortality rates and long-term complications in children with infectious brainstem encephalitis: A study of sixteen casesPLOS ONE

Dear Dr. Shen,

Thank you for submitting your manuscript to PLOS ONE. After careful consideration, we feel that it has merit but does not fully meet PLOS ONE’s publication criteria as it currently stands. Therefore, we invite you to submit a revised version of the manuscript that addresses the points raised during the review process.

We look forward to receiving your revised manuscript.

Kind regards,

Benjamin M. Liu, MBBS, PhD, D(ABMM), MB(ASCP)

Academic Editor

PLOS ONE

Journal Requirements:

 “This study was supported by The National Key Research and Development Program of China (2022YFC2704901), the Project of Invigorating Health Care through Science, Technology and Education (Nantong Municipal Key Medical Discipline), and Beijing Natural Science Foundation (L242052).”

6. Please provide a complete Data Availability Statement in the submission form, ensuring you include all necessary access information or a reason for why you are unable to make your data freely accessible. If your research concerns only data provided within your submission, please write "All data are in the manuscript and/or supporting information files" as your Data Availability Statement.

Additional Editor Comments:

The Introduction section should include introduction on diagnostic tools for BE. Multiplex PCR panel, e.g., BioFire Meningitis/Encephalitis panel (PMID: 38042947) and mNGS (PMID: 39221481), as well as potential pathogens, e.g., HSV, HEV (PMID: 28031445), human parechovirus (PMID: 31283959), should be introduced. More references should be cited, with the above-mentioned references as examples (citing is optional)

Reviewers' comments:

Reviewer's Responses to Questions

**Comments to the Author**

1. Is the manuscript technically sound, and do the data support the conclusions?

Reviewer #1: Yes

Reviewer #2: Yes

2. Has the statistical analysis been performed appropriately and rigorously? 

Reviewer #1: Yes

Reviewer #2: Yes

3. Have the authors made all data underlying the findings in their manuscript fully available?

Reviewer #1: Yes

Reviewer #2: Yes

4. Is the manuscript presented in an intelligible fashion and written in standard English?

Reviewer #1: Yes

Reviewer #2: Yes

5. Review Comments to the Author

Reviewer #1: A retrospective study was conducted on pediatric brainstem encephalitis (BE) cases at a National Children's Medical Center from 2015 to 2024.

I would expect the following from a retrospective study:

1. BE is a rare disease, and I would expect an analysis of its frequency, demographic patterns, and risk factors over a long period.

2. I also expect clinical manifestations and diagnosis to highlight common symptoms and pathogens causing BE, providing clues for diagnostic protocols and management.

In this study, the authors reported key symptoms such as fever, consciousness disorders, and seizures. However, I suggest including additional data such as demographic details, personal history (e.g., autoimmunity, immune checkpoint inhibitor exposure), immune response markers, brainstem-specific signs and symptoms for a more detailed analysis. A risk factor analysis should also be included.

The authors also reported the pathogens associated with BE, cranial imaging, and electroencephalogram findings, as well as the outcomes of several treatments. However, the data could be more comprehensive by including MRI findings, CSF analysis, oncological features, and antibody-specific associations. While the authors mentioned that CSF data is unavailable, its inclusion would greatly enhance the study's value.

Overall, I consider this study preliminary. More data could be added as suggested above, along with deeper analyses. The illustrations can also be improved

Reviewer #2: Brainstem encephalitis is defined as a disorder characterized by myoclonus, ataxia, nystagmus, kinetic nerve palsy, and bulbar palsy and various combinations of these disorders, whether or not confirmed by neuroimaging. They are often easily overlooked by physicians and parents, and this may lead to misdiagnosis. If the medullary vital centers are involved, the condition can deteriorate rapidly and even lead to death. Early diagnosis of the presence of a viral infection and early intervention are critical.

The paper analyzes the prevalence of brainstem encephalitis among 30 million outpatients and 500,000 inpatients from real-world clinical data over the past decade, and analyzes the etiology, clinical manifestations, imaging, diagnosis and treatment, and prognostic follow-up in detail.

The paper is concise, logical, strongly argued, with accurate data and clear conclusions.

For clinicians, researchers and patients, it is a rare first-hand data, which is of great reference value. Especially for this kind of disease with atypical symptoms, easy to misdiagnose, low population prevalence relative to common diseases, and poor prognosis once the disease develops, timely and adequate retrospective study is very necessary. This is also reflected in the analysis of infectious etiologies, including, but not limited to, enteroviral involvement in neurological infections, etc., which are well worth following up with studies of pathogenic mechanisms to facilitate the development of effective drugs and vaccines.

Finally, I would like to make a few suggestions without prejudice to the evaluation of the full paper.

The baseline characteristics of patients (Line 105), clinical manifestations (Line 126), etiology (Line 139), prognosis (Line 177), and follow-up (Line 192) should be matched with corresponding figures and tables, rather than just textual descriptions, which may be more convincing and convenient for the readers to understand the data information intuitively, especially the baseline characteristics of the cohort and the survival curve.

6. PLOS authors have the option to publish the peer review history of their article (what does this mean?). If published, this will include your full peer review and any attached files.

Reviewer #1: No

Reviewer #2: No

---

## [Author Response · Author response to Decision Letter 0]

15 Jan 2025

Author’s response to reviewers

Reviewer #1: 

A retrospective study was conducted on pediatric brainstem encephalitis (BE) cases at a National Children's Medical Center from 2015 to 2024.

I would expect the following from a retrospective study:

1. BE is a rare disease, and I would expect an analysis of its frequency, demographic patterns, and risk factors over a long period.

Author’s response:

Thank you for your valuable suggestions. Infective BE in children is indeed a rare clinical condition, yet its severity necessitates heightened vigilance among clinicians. To date, no incidence data could be found in the literatures. Given that patients at our Children's National Medical Center are typically complex or critically ill, this patient population may not accurately reflect the overall prevalence of infectious BE. In this paper, we provide annual case counts to illustrate the potential frequency of this disease. Specifically, among 3 million children visiting the outpatient department and 500,000 hospitalized children, there were 16 and 13 cases, respectively. In the revised manuscript, we have integrated these two sets of data for a comprehensive presentation. And we have incorporated Table 1 into the revised draft to offer a comprehensive overview of the clinical information pertaining to these 16 cases.

In response to the reviewer's suggestion, we have incorporated the age range into the demographic characteristics section of the revised manuscript. Concerning the description of risk factors, given that only one out of the 16 patients had an underlying condition (congenital heart disease), demographic variables such as age and sex did not appear to be indicative of risk factors, likely due to the limited sample size. In the discussion section, we have provided additional clarification regarding the risk factors.

2. I also expect clinical manifestations and diagnosis to highlight common symptoms and pathogens causing BE, providing clues for diagnostic protocols and management.

In this study, the authors reported key symptoms such as fever, consciousness disorders, and seizures. However, I suggest including additional data such as demographic details, personal history (e.g., autoimmunity, immune checkpoint inhibitor exposure), immune response markers, brainstem-specific signs and symptoms for a more detailed analysis. A risk factor analysis should also be included.

The authors also reported the pathogens associated with BE, cranial imaging, and electroencephalogram findings, as well as the outcomes of several treatments. However, the data could be more comprehensive by including MRI findings, CSF analysis, oncological features, and antibody-specific associations. While the authors mentioned that CSF data is unavailable, its inclusion would greatly enhance the study's value.

Overall, I consider this study preliminary. More data could be added as suggested above, along with deeper analyses. The illustrations can also be improved.

Author’s response:

We would like to express our gratitude to the reviewers for acknowledging the clinical and scientific significance of the manuscript, as well as for their insightful recommendations. In response, we have augmented Table 1 with detailed clinical information pertaining to the 16 patients, including cerebrospinal fluid test results.

Reviewer #2: 

Brainstem encephalitis is defined as a disorder characterized by myoclonus, ataxia, nystagmus, kinetic nerve palsy, and bulbar palsy and various combinations of these disorders, whether or not confirmed by neuroimaging. They are often easily overlooked by physicians and parents, and this may lead to misdiagnosis. If the medullary vital centers are involved, the condition can deteriorate rapidly and even lead to death. Early diagnosis of the presence of a viral infection and early intervention are critical.

The paper analyzes the prevalence of brainstem encephalitis among 30 million outpatients and 500,000 inpatients from real-world clinical data over the past decade, and analyzes the etiology, clinical manifestations, imaging, diagnosis and treatment, and prognostic follow-up in detail.

The paper is concise, logical, strongly argued, with accurate data and clear conclusions.

For clinicians, researchers and patients, it is a rare first-hand data, which is of great reference value. Especially for this kind of disease with atypical symptoms, easy to misdiagnose, low population prevalence relative to common diseases, and poor prognosis once the disease develops, timely and adequate retrospective study is very necessary. This is also reflected in the analysis of infectious etiologies, including, but not limited to, enteroviral involvement in neurological infections, etc., which are well worth following up with studies of pathogenic mechanisms to facilitate the development of effective drugs and vaccines.

Finally, I would like to make a few suggestions without prejudice to the evaluation of the full paper.

The baseline characteristics of patients (Line 105), clinical manifestations (Line 126), etiology (Line 139), prognosis (Line 177), and follow-up (Line 192) should be matched with corresponding figures and tables, rather than just textual descriptions, which may be more convincing and convenient for the readers to understand the data information intuitively, especially the baseline characteristics of the cohort and the survival curve.

Author’s response:

Thank the reviewers for their insightful and professional comments. Given the rapid progression, serious consequences, and lack of clinical specificity associated with infectious brainstem encephalitis, timely diagnosis often necessitates a combination of clinical vigilance and laboratory examinations, including neuroimaging, to facilitate prompt and targeted treatment measures, thereby preventing adverse outcomes. Based on the reviewers' suggestions, we have supplemented the detailed information of these 16 patients in a more intuitive format as presented in Table 1. We anticipate that the successful publication of our manuscript will contribute to a deeper understanding of infectious BE among clinicians and benefit patient care.

Author’s response to editors

Author’s response: Yes.

Author’s response: Yes.

Author’s response: Yes.

Journal Requirements:

1.Please ensure that your manuscript meets PLOS ONE's style requirements, including those for file naming. 

Author’s response: Yes.

2. Please provide additional details regarding participant consent. 

Author’s response: Yes.

3.. Please note that funding information should not appear in any section or other areas of your manuscript.

Author’s response: Yes.

Author’s response: Yes, we made changes.

 “This study was supported by The National Key Research and Development Program of China (2022YFC2704901), the Project of Invigorating Health Care through Science, Technology and Education (Nantong Municipal Key Medical Discipline), and Beijing Natural Science Foundation (L242052).”

Author’s response: Yes. we amended the Role of Funder statement in the manuscript and cover letter. 

6. Please provide a complete Data Availability Statement in the submission form, ensuring you include all necessary access information or a reason for why you are unable to make your data freely accessible. If your research concerns only data provided within your submission, please write "All data are in the manuscript and/or supporting information files" as your Data Availability Statement.

Author’s response: Yes. we added "All data are in the manuscript".

7. Please include captions for your Supporting Information files at the end of your manuscript, and update any in-text citations to match accordingly. 

Author’s response: Yes. 

Additional Editor Comments:

The Introduction section should include introduction on diagnostic tools for BE. Multiplex PCR panel, e.g., BioFire Meningitis/Encephalitis panel (PMID: 38042947) and mNGS (PMID: 39221481), as well as potential pathogens, e.g., HSV, HEV (PMID: 28031445), human parechovirus (PMID: 31283959), should be introduced.

Author’s response: Yes. we cited more references.

---

## [Editor Report · Decision Letter 1]

22 Jan 2025

High mortality rates and long-term complications in children with infectious brainstem encephalitis: A study of sixteen cases

PONE-D-24-57941R1

Dear Dr. Shen,

We’re pleased to inform you that your manuscript has been judged scientifically suitable for publication and will be formally accepted for publication once it meets all outstanding technical requirements.

Kind regards,

Benjamin M. Liu, MBBS, PhD, D(ABMM), MB(ASCP)

Academic Editor

PLOS ONE
---

## [Editor Report · Acceptance letter]

24 Jan 2025

PONE-D-24-57941R1 

PLOS ONE

Dear Dr. Shen, 

I'm pleased to inform you that your manuscript has been deemed suitable for publication in PLOS ONE. Congratulations! Your manuscript is now being handed over to our production team.

Kind regards, 

on behalf of

Dr. Benjamin M. Liu 

Academic Editor

PLOS ONE